

# Spatial and temporal features of snow water equivalent across a headwater catchment in the Sierra Nevada, USA

Ernesto Trujillo[1, 2*], Andrew Hedrick[2], and Danny Marks[2]

1. Department of Geosciences, Boise State University, Boise, ID, USA
2. USDA-ARS Northwest Watershed Research Center, Boise, ID, USA

* Now at HDR, Inc., Sacramento, California, USA.

*Correspondence to*: Andrew Hedrick (Andrew.Hedrick@usda.gov)

**Abstract.** Advancements in remote sensing of snow (e.g., lidar) have allowed for the characterization of mountain snowpacks at higher spatial resolutions (< 10 m) and higher vertical accuracy (< 20 cm) than previously available, which can cover entire catchments repeatedly throughout the snow season. Here, we use distributed snow water equivalent (SWE) over the Tuolumne River Basin in California, USA, from lidar snow depths combined with energy/mass balance modeling of the Airborne Snow Observatory (ASO) program for the period 2013-2017 (48 flight-dates, 50-m resolution) to characterize the spatial and temporal variations in SWE distribution in this headwater catchment. Peak basin snow volume storage ranged from 142 M m$^3$ (i.e., $10^6$ m$^3$) in 2015 to 1467 M m$^3$ in 2017, covering one of the widest ranges in recorded history. Basin SWE empirical distributions vary between unimodal and bimodal distributions earlier in the season to decaying distributions later in the ablation season. Snow storage peaks at mid-elevations between 2750-3250 m, which is a consequence of increases in SWE with elevation and basin hypsometry. The date of peak SWE varies by several weeks across the watershed and between years, according to the combination of accumulation and melt patterns partially explained by elevation and aspect. This variation in peak SWE timing leads to underestimations if a single date is used to uniformly characterize the basin's peak SWE. These results illustrate how understanding SWE spatiotemporal dynamics can improve the understanding of where the snow is and when it melts, support satellite mission planning, and enhance ground survey design and planning.



## 1    Introduction

Snowpack storage in headwater catchments is a critical water resource across the globe (Viviroli et al., 2007). In the high elevation regions of the western United States (US), precipitation in the form of snow can account for

as much as 50% - 70% of the annual precipitation, with the highest percentages in the Sierra Nevada (67%), northwestern Wyoming (64%), Colorado (63%), and Idaho/western Montana (62%) (Serreze et al., 1999). This snow is then stored in the mountains throughout the winter and is later released during the spring and summer melt. Because of this, the symbolic term "water towers" has been coined to refer to mountain snowpacks in these snow dominated regions (Liniger et al., 1998; Bandyopadhyay et al., 1997; Viviroli et al., 2007).

Understanding snowpack dynamics across headwater catchments is of particular importance for understanding how inter- and intra-annual variations in snowpack accumulation and melt affect ecological function (Knowles et al., 2018; Trujillo et al., 2012), water supply (Mankin et al., 2015; Barnett et al., 2005; Viviroli et al., 2007), and recreation (Burakowski and Magnusson, 2012; Sturm et al., 2017). Knowledge of where the snow is, when it melts, and snowpack storage allows water managers to make decisions on reservoir water storage and water

releases, allocations for hydroelectric energy generation and ecological flow requirements. Similarly, the optimal design of measurement campaigns largely depends on the snow distribution at a given time, particularly when water forecasting agencies depend on this information to forecast available snow water storage (Fleming and Goodbody, 2019; Garen, 1992).

Maximum annual snow accumulation, or peak snow water equivalent (SWE), is useful for determining the

amount of snowmelt water that will be available during the spring and summer. However, the stochastic nature of the date of peak SWE across landscapes and snow regimes (Trujillo and Molotch, 2014) make the determination of peak SWE challenging across large headwater basins. Manual SWE measurements on nominal survey dates (e.g., bi-weekly or monthly) near or around the date of peak SWE have often been used instead. Following these practical considerations, April 1 SWE has often been used as an approximate peak SWE metric

useful when estimating seasonal flows, providing better empirical estimates when compared to earlier or later monthly or biweekly SWE measurements  (Bohr and Aguado, 2001). Bohr and Aguado (2001) found that April 1 SWE underestimated peak SWE by between 9-19 % at 92 snow pillow stations between 1981 and 1996 in the Wasatch and Uinta Mountains, the Rocky Mountains and the Idaho Batholith. Trujillo and Molotch (2014) analyzed daily SWE from over 700 snow pillow stations across the western US between 1980 and 2009,

showing that peak SWE can occur over a wide period, with the majority of observations between early February and late May for snowpacks with peak SWE greater than 250 mm. In the same study, Trujillo and Molotch (2014) demonstrated complex relationships between peak SWE, the date of peak SWE and other snowpack metrics such as the ablation rates and the date of snow disappearance, illustrating dynamics in the snow accumulation and melt cycles across the western US that are consistent with the climatology of the regions.

Despite this, April 1 SWE continues to be used as a useful metric to characterize snow storage trends (e.g., Musselman et al., 2021) and "snow drought" (e.g., Pederson et al., 2011; Harpold et al., 2017), a concept that is gaining traction to characterize water stress in years with reduced snowpack storage. Hatchett and McEnvoy (2018) further highlighted possible issues of using April 1 SWE when defining snow drought, as the large variability of the date of peak SWE across landscapes and regions could lead to mischaracterizations of peak



snow storage. Similar watershed scale analyses of these complex dynamics in the snow accumulation and melt cycles and the timing of peak snow storage have been limited by the small number of snow pillow stations and/or snow courses within watersheds and the limited spatially distributed SWE datasets throughout the snow season.

The study of the spatial and temporal dynamics of snow distribution is often approached through the analysis of

snow depth (Trujillo et al., 2009, 2007; e.g., Deems et al., 2006; Grünewald et al., 2010, 2014; Schirmer et al., 2011; Schirmer and Lehning, 2011; Schirmer and Pomeroy, 2020) and SWE (Elder et al., 1998, 1991; Trujillo and Molotch, 2014; Garen, 1992; e.g., Anderton et al., 2004; Rice et al., 2011; McCartney et al., 2006). Snow depth has often been favored because of the ease of obtaining direct measurements through snow probing or directly inferred from snow-on and snow-off topographic surveys (e.g., lidar, terrestrial laser scanning (TLS),

and Structure from Motion (SfM)). These latter remote sensing techniques also offer unrivaled horizontal spatial resolutions (centimeter to meters) and vertical accuracy (centimeters), and can provide spatial coverage from slope to watershed scales in mountainous terrain (Prokop et al., 2008; Buhler et al., 2016; Deems et al., 2013; Harder et al., 2016; Walker et al., 2020; Pelto et al., 2019). Analyzing SWE offers the benefit of providing information about the water mass in the snowpack, which is particularly useful for the study of the water cycle

and hydrological modeling applications. However, direct SWE measurements are more resource-intensive and therefore are more scarce (e.g., snow pillows, snow cores and snow pits), and satellite remotely-sensed distributed datasets lack the spatial resolution (e.g., 8-70 km from the Scanning Multichannel Microwave Radiometer (SMMR), the Special Sensor Microwave/Imager (SSM/I) and the Advanced Microwave Scanning Radiometer for the NASA Earth Observing System (AMSR-E) (Foster et al., 2005)), and accuracy required in

complex, mountainous, and forested terrain (Chang et al., 1996; Dong et al., 2005; Foster et al., 2005). Furthermore, point datasets of either snow depth or SWE provide limited spatial information due to the spatial discontinuity and small support (i.e., spatial representativeness) of the measurements, while interpolation techniques fail to capture the true spatial variability of snow distribution (Trujillo and Lehning, 2015; Erxleben et al., 2002; Dickinson and Whiteley, 1972). Despite this, many of the studies mentioned above have relied on

interpolated datasets or the combination of interpolated snow depth and snow density, partly because such methods provided the "best case scenario" for the available resources and information at hand.

A more recent approach to obtaining distributed SWE information throughout the snow season consists of combining lidar snow depths and modeled snow densities, as proposed and implemented by the NASA JPL Airborne Snow Observatory (ASO) and the USDA Agricultural Research Service (ARS) (Painter et al., 2016).

In this approach, repeated lidar snow depth surveys covering entire basin domains are combined with spatially distributed bulk snow density estimates from the physically based snow model iSnobal (Marks et al., 1999) to provide SWE estimates at 50-m resolutions for headwater catchments larger than 1000 km$^2$. The iSnobal modeled snow depths are also updated with the lidar snow depths from each ASO survey following the direct insertion approach described in Hedrick et al. (2018b). Since 2013, the approach has been successfully

implemented in the headwaters of the upper Tuolumne River Basin in the California Sierra Nevada (Painter et al., 2016), and since 2019 has been expanded to dozens of headwater catchments across the western US through the efforts of the public benefit corporation ASO, Inc. These ASO datasets offer the advantage of accurate high resolution snow depth data (e.g., 3 m resampled to match 50-m model resolutions) and model results



constrained by the lidar surveys (Hedrick et al., 2018b), offering full spatial coverage at multiple dates each
season (weekly-monthly).

Here, we take advantage of the NASA-JPL ASO dataset for the upper Tuolumne River Basin for water years
(WYs, WY for a singular year hereinafter) 2013-2017 to analyze the spatial and temporal dynamics of snow
accumulation and melt in a headwater catchment at combined scales, resolutions, and temporal coverage
unavailable prior to the development of this integrated dataset. The analyses in this study provide a
comprehensive look across both space and time at snowpack storage around the dates of peak accumulation and
the subsequent snowmelt season across the watershed domain, with emphasis on the transitional characteristics
of snowpack storage and the implications of these results for both research and practical applications. We
perform a series of analyses to address the following research questions:

(a)   What is the range of variability in basin snowpack storage for years with very different snow seasons

115            across a large headwater catchment? And how do statistical properties (i.e., empirical probability

distributions) of SWE across a headwater catchment vary around the dates of peak SWE and the

subsequent snowmelt season, and between very different snow seasons?

(b)   How does snowpack storage vary across elevations from peak conditions onwards, and what are the

differences in snowmelt dynamics across elevations?

(c)   What is the range of variability in the date of peak SWE across a large headwater catchment and how

does this variability relate to topographic characteristics (i.e., elevation and aspect)? Then, how does

peak SWE variability relate to topographic characteristics (i.e., elevation and aspect), and what are the

implications of only obtaining a single lidar survey around the date of basin peak snowpack storage?

Results are discussed in the context of previous research, with an emphasis on the relationships between spatial
and temporal snowpack dynamics, and basin characteristics across years with very different snow seasons. The
extreme conditions observed during the study period provide relevant insight on the effects of the natural
variability in the weather patterns of the Sierra Nevada region on water resources from snowpack storage.
Although previous studies have provided relevant insights regarding some of these questions, that understanding
is still relatively vague, in great part because of the limitations in data availability to perform more
comprehensive analyses and over larger spatial scales. These analyses offer a first look at the types of analyses
possible as similar datasets become available for many regions across the globe, and as new remote sensing
technologies, either from airborne or satellite platforms, extend the existing available technologies for
measuring snow depth and SWE.




## 2    Data and Methods

The Tuolumne River Basin is a headwater catchment in the California Sierra Nevada (Figure 1) with elevations ranging between 1150 m and 3999 m, and with a contributing area of 1180 km$^2$ to the Hetch Hetchy Reservoir (Hedrick et al., 2018b). Treeline in the watershed is located at around 2900 m, with land cover dominated by

conifer forest below this elevation, and shrubs and exposed granite bedrock above. The watershed is dominated by north-west and south-east aspects (Figure 2a and b) consistent with the predominant orientation of the river network    (generated    using    TauDEM    from    a    10-m    digital    elevation    model    (DEM); http://hydrology.usu.edu/taudem/taudem5/index.html,). Elevations above 2000 m (90% of the area, Figure 2c and d) are snow-dominated (>~70 % of annual precipitation falling as snow (Hedrick et al., 2018b)), while the

lower elevations are rain dominated and/or in the rain/snow transition zone, although such elevations are variable between storms (Lundquist et al., 2016). The contributing area is dominated by mid-elevations between 2500 m and 3250 m (64 % of the basin) (Figure 2c and d), with 26 % below 2500 m and 10 % above 3250 m.

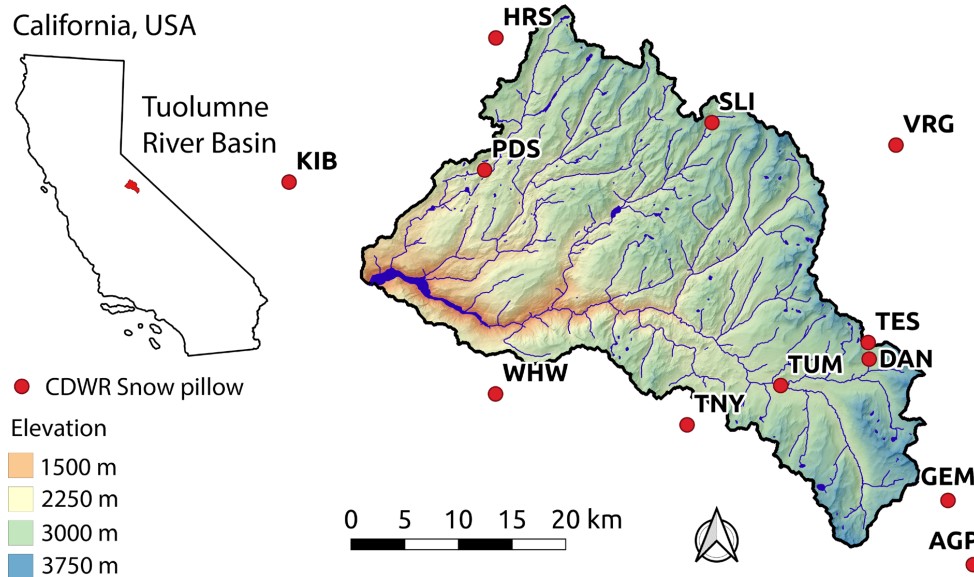

**Figure 1. Location of the Tuolumne River Basin in the state of California, and contributing area to the Hetch Hetchy Reservoir with elevations, river network, and snow pillow stations from the California Department of Water Resources (CDWR) and the Natural Resources Conservation Service (NRCS) SNOTEL network.**

The dataset used here is the product of ASO lidar snow depths (3-m resolution snow depths resampled to 50-m) and iSnobal modeled snow densities at matching 50-m resolution for WYs 2013-2017 (Trujillo et al., 2025). The methodology to produce the 50-m SWE dataset is fully described in Hedrick et al. (2018b, 2020) and were



generated through a reproducible framework (Hedrick et al., 2018a). The process of deriving hourly 50-m forcing grids from station measurements of air temperature, relative humidity, wind speed, precipitation, and incoming solar irradiance is described in full by Havens et al. (2017). For the extreme snow year 2017 during which some weather stations were buried, the model was forced using weather and precipitation data from the High Resolution Rapid Refresh atmospheric model (HRRR, Benjamin et al., 2016) following the methodologies laid out by Havens et al. (2019) and Meyer et al. (2023). Vegetation data from the National Land Cover Database (https://www.usgs.gov/centers/eros/science/national-land-cover-database) was used to estimate canopy effects on net radiation and turbulent fluxes following the methodologies described in Link and Marks (1999b, a) and Marks et al. (2008) implemented in iSnobal.

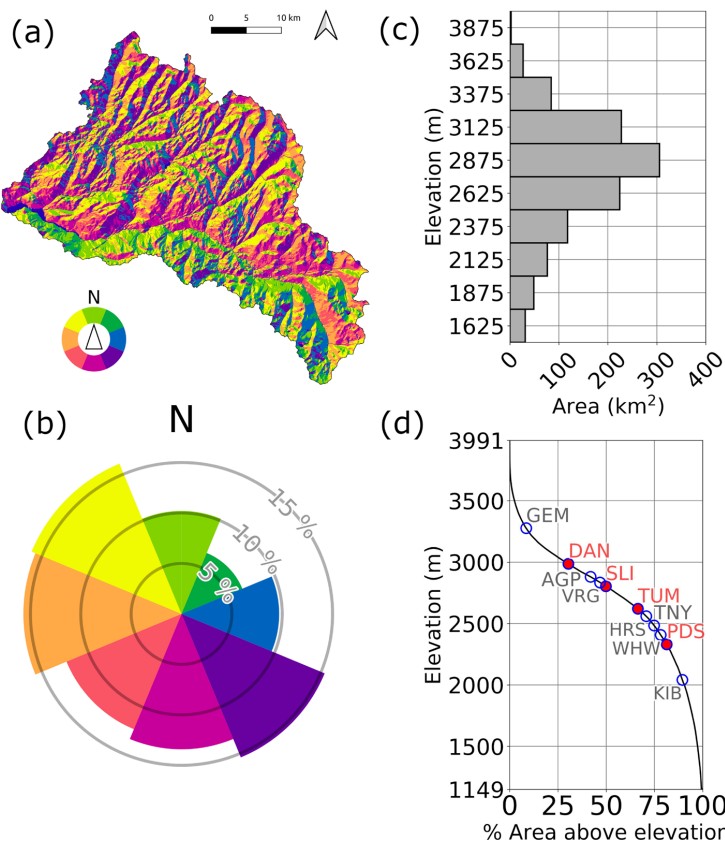

**Figure 2. (a) Aspect, (b) Aspect distribution as a percentage of the basin area (45º bins), (c) basin area distribution across elevation (250-m bins) for the Tuolumne River Basin, and (d) hypsometric curve with elevation of snow pillow stations, and with stations within the watershed boundaries highlighted in red. Aspect coloring scheme following Brewer & Marlow (1993).**

Lidar snow depths from ASO (Painter et al., 2016) were used to periodically update the model snow depth state variable using the direct insertion approach (Hedrick et al., 2018b), which leaves the remaining model state variables of density, snow temperature, and liquid water content unchanged. This approach allows for the



correction of the snow distribution by adding spatial variability to the snow fields in the model. Furthermore, although the effects of mass transport due to avalanche or wind redistribution are not explicitly addressed in the model, the direct insertion of the lidar snow depths allows for the correction of where these deposits are located

with each update. The uncertainty associated with the 50-m lidar snow depth ASO product has been estimated to be on the order of 0.08 m (Painter et al., 2016).

In iSnobal, the density of newly fallen snow was estimated using storm dew point temperatures, after which bulk snowpack density was tracked using snowpack temperature gradients, overburden pressure, and the presence of liquid water within the ice matrix. The snow density model implemented in iSnobal follows the

approach by Anderson (1968, 1976), which was based on the field and cold room measurements by Yosida (1963, 1958), Mellor (1964) and Kojima (1967). These same equations from Anderson (1976, 1968) were later updated by Oleson et al. (2013) within the Community Land Model (CLM), and were further adjusted to estimate bulk snow density before being incorporated into iSnobal. Comparisons at the point scale of simulated to in-situ bulk density at 11 validation sites in Idaho and California during the density model development and

implementation showed Nash-Sutcliffe coefficients of 0.83 +/- 0.08, mean bias error of +13 +/- 14 kg m$^{-3}$ and root-mean square difference of 40 +/- 13 kg m$^{-3}$. Evaluations of model densities at the gridded spatial scales (e.g., 50 m) are more challenging and have unquantifiable added uncertainties due to the discrepancy of scales between available measurements (e.g., snow courses, snow pillows and snow pits) and the spatial grid cell footprint and thus have not been performed. Note here that measurements from snow courses are often reported

as the average of several snow core measurements along transects that can be more than a kilometer long (e.g., from the California Department of Water Resources (CDWR) through https://cdec.water.ca.gov/). Snow densities are often derived from these reported mean SWE and mean snow depth values for these transect/courses, resulting in uncertainties that cannot be directly quantified. Additionally, uncertainties are also associated with direct local measurements of snow density. For example, bulk densities from snow pit profiles

can carry 10+% uncertainties and are seldom available in practical settings (Proksch et al., 2016). Similarly, snow depth sensors at snow pillow stations are not always located directly over the snow pillow, leading to uncertainties that are difficult to address, with errors in derived snow densities that become more pronounced when the snowpack is shallower (e.g., early or late snow season). During the study period of 2013 – 2017, this location offset between sensors was confirmed at several snow pillow stations within the rectangular Tuolumne

modeling domain, including a station within the Tuolumne Basin (i.e., Dana Meadows – DAN, Figure 1).

The resulting dataset used in this study consists of distributed SWE covering the entire watershed at 50-m resolutions. Here we use a total of 48 lidar survey dates during the period WYs 2013-2017, with flight intervals (days between flights) ranging from 4 to 35 days, and with 6 to 13 flights per year (Table 1). The survey dates generally capture snow conditions around or just prior to peak snowpack storage and throughout the subsequent

snowmelt seasons. In WY 2017, flights commenced in January and continued well into the accumulation season (Figure 3 and Table 1). The study period provides the unique opportunity to analyze some of the most extreme water years on record, with the extreme drought of WY 2015, followed by the near average WY 2016, and culminating with WY 2017, one of the snowiest water years on record (Ullrich et al., 2018; Wang et al., 2017).

Daily SWE from snow pillow stations from the California Department of Water Resources (CDWR) and the

Natural Resources Conservation Service (NRCS) SNOTEL network (Figure 1) are also used to illustrate the daily progression of the snow seasons during the study period. The snow pillow stations are located between





2000 m and 3300 m, although only four are located within the watershed boundaries and cover a narrower elevation range between 2350 m and 3000 m (Figure 2d).

220 **Table 1. ASO Survey dataset information.**

| Water Year | No. of flights | Interval | Period (month/day) |
|------------|----------------|----------|--------------------|
| 2013 | 6 | 4 - 26 days | 04/03 – 06/08 |
| 2014 | 11 | 4 - 15 days | 03/23 – 06/05 |
| 2015 | 10 | 4 - 27 days | 02/18 – 06/08 |
| 2016 | 13 | 5 - 18 days | 03/26 – 07/08 |
| 2017 | 8 | 8 - 35 days | 01/29 – 08/16 |

In order to address the research questions presented in the Introduction, the following analyses are performed:

 i. *Basin SWE s*torage: The total volume of water stored in the form of snow within the basin boundaries was estimated as:

225
$$Basin\ SWE\ Vol. = \sum_{i=1}^{N} SWE_i \times A_i$$

Equation 1

where $i$ is the grid-cell number, $N$ is the total number of grid-cells within the basin boundaries, and $SWE_i$ and $A_i$ are the corresponding grid-cell SWE and area (50 m by 50 m), respectively.

 ii. *SWE empirical probability distributions*: Empirical probability distribution functions of SWE

230 were derived for each of the 48 flight dates, and a representative selection of six per year are used in the analyses below.

 iii. *Elevational variations in snow accumulation*: Two different variables are analyzed here: (a) the volume of SWE stored in 250-m elevation bands within the basin boundaries, and (b) mean SWE within corresponding 250-m elevation bands. The former is a combination of SWE and area

235 within each band following the same formulation as in Equation 1, with an intrinsic influence of the hypsometry of the watershed (Figure 2), while the latter represents the average snow accumulation across elevation as:

$$Elev.\ Band\ Mean\ SWE\ = \frac{1}{N} \sum_{i=1}^{N} SWE_i$$

Equation 2

240 where $i$ is the grid-cell number, $N$ is the total number of grid-cells within each elevation band and the basin boundaries, and $SWE_i$ is the corresponding grid-cell SWE.



iv. *Spatially resolved date of peak SWE across the basin*: In these analyses, each grid-cell was treated individually such that a peak date was determined for each grid-cell, providing a spatially resolved image of peak SWE date for each WY. The resulting spatial characterization of the timing of peak SWE extends previous point/in-situ characterizations (e.g., Trujillo and Molotch, 2014) to a large headwater catchment spatial domain.

v. *Spatially resolved peak SWE*: A consequence of the spatial variability in the timing of peak is that local peak SWE (e.g., at a point in the basin) can be larger than SWE at the timing of basin peak SWE storage. Therefore, spatially resolved peak SWE maps were estimated for each year considering each grid-cell in the watershed individually to determine peak SWE. The difference between these spatially distributed peak SWE maps and SWE at the timing of basin peak SWE storage can be used to quantify the spatial underestimation of peak SWE over a headwater catchment when the standard concept of assuming a uniform single peak SWE date (e.g., April 1) over a spatial domain is applied.

Throughout this article, the term "snow accumulations" is used to refer to the amount of SWE accumulated on the landscape. The terms "peak snow accumulations" and "peak SWE" are used interchangeably to refer to the maximum amount of snowpack SWE in snow season. Peak snow conditions at both basin and local scales (i.e., basin peak SWE and spatially resolved peak SWE) are determined from the 48 survey dates, with 6-13 survey dates per year (Table 1). Lastly, ablation and redistribution processes, such as snow sublimation and wind redistribution, are ubiquitous throughout the snow season, and are not restricted to the spring and summer snowmelt season. Similarly, the processes through which SWE is added to the snowpack from snow precipitation can also be present throughout the snow season.

## 3    Results

### 3.1    Basin SWE storage

The period of WYs 2013-2017 provided a wide range of snow conditions (Figure 3), with the lowest snow accumulations in WY 2015, and the largest snow accumulations in WY 2017, a year that delivered record snowfall across the western US. Peak basin SWE storage (i.e., total volume of water stored as snowpack within the basin boundaries, and calculated following Equation 1) in the Tuolumne River basin ranged between 142 M $m^3$ on March 3, 2015, and 1467 M $m^3$ on April 1, 2017 (Figure 3a). SWE measurements from snow pillow stations in and around the basin (Figure 3b) confirm that peak snow accumulations occurred near or after the time of the first ASO flight in most years. Peak basin SWE storage was likely reached just prior to the first ASO flight during WY 2013, and between the two ASO flights of April 1 and May 2 in WY 2017, following precipitation events that occurred after the April 1 flight.



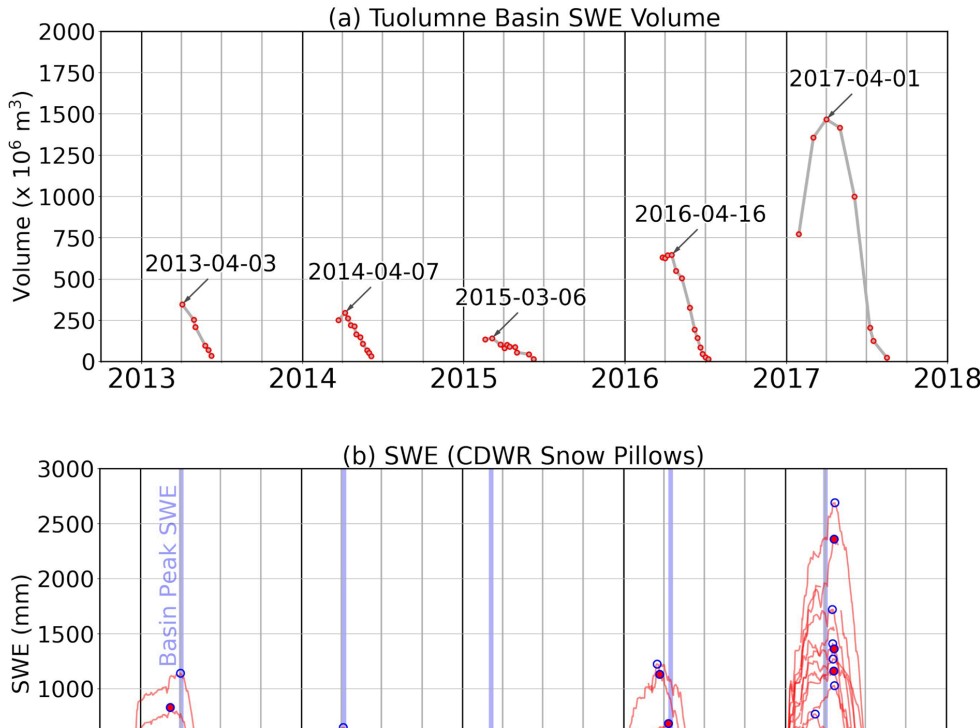


**Figure 3. (a) Basin SWE storage calculated following Equation 1 and dates of peak storage for the study period WYs 2013-2017. Dates of lidar flights are denoted by the red circle markers. Straight lines are used to connect results between each survey date. (b) Daily SWE from the CDWR and SNOTEL snow pillow stations (red lines, each red line corresponding to an individual snow pillow station), with the dates**

**of basin peak SWE (from (a)) highlighted by the vertical blue lines, and snow pillow peak SWE highlighted by the blue circle markers (markers for stations within the basin boundaries are filled in with solid red).**

### 3.2 SWE Empirical Distributions

Owing to the range of variability in annual SWE totals, basin SWE empirical probability distribution functions can be analyzed in a way that was not possible prior to the NASA-JPL ASO measurement and USDA-ARS iSnobal modeling program (Figure 4-Figure 6). SWE empirical distributions throughout dry (Figure 4) and average water years (Figure 5) are unimodal and truncated at zero SWE around peak accumulation. In the case of SWE (and snow depth), values have a zero lower bound limiting the distributions to the positive range $[0, \infty)$

(i.e., truncated). Once the melting season is established, the distributions follow a decaying function that can be conceptualized as if the distribution moved left as the seasons progressed and both SWE storage and snow-



covered area (SCA) decrease because of melt. On the other hand, during the extreme snow conditions in WY 2017, storm events in March and April transformed the empirical distribution from a unimodal to a bimodal distribution with a second distinct peak for SWE accumulations larger than 2000 mm (Figure 6). This bimodal

distribution pattern is explained by the distribution of snowfall during the snow storms of March and April, which delivered large amounts of snow at higher elevations. These deposition patterns are evidenced in SWE difference maps between the March 3, April 1, and May 2 flights in 2017 (not shown). Despite this bimodal pattern, the distribution returns to the decaying function observed during drier years in the melting season once SWE storage and SCA decrease (Figure 6).


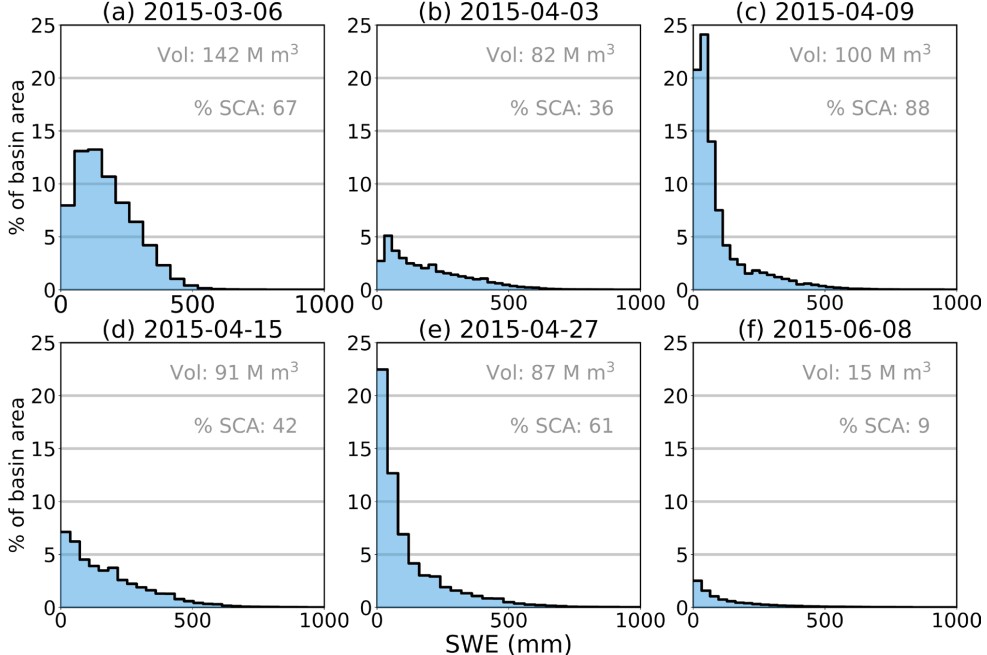

**Figure 4. Histograms of SWE, basin SWE storage and basin percent of snow-covered area (SCA) for six flight dates in the drought WY 2015.**



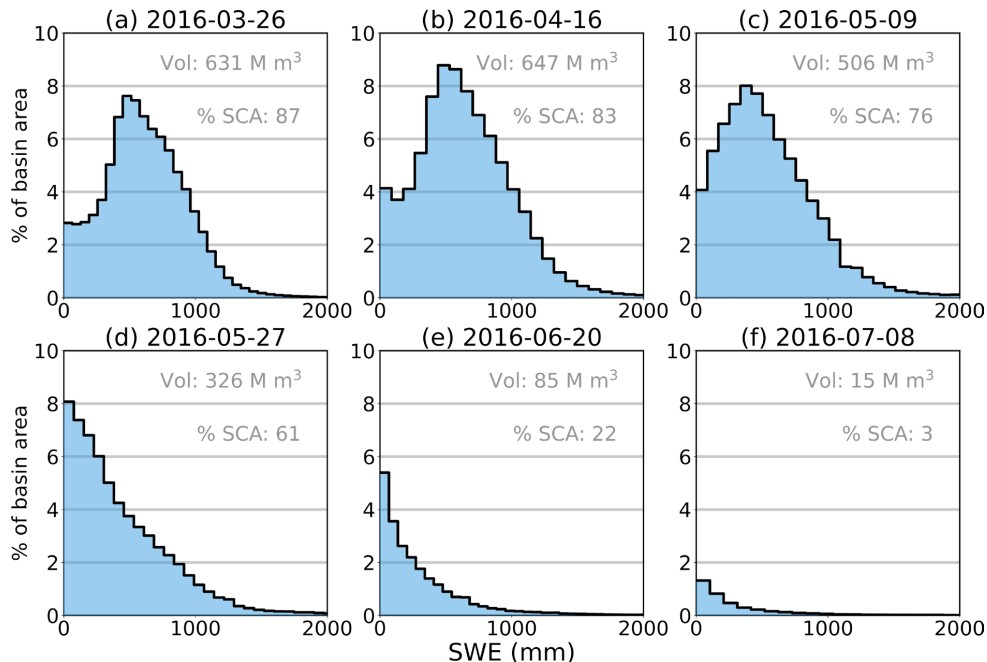

**Figure 5. Same as Figure 4 but for the near-average WY 2016.**

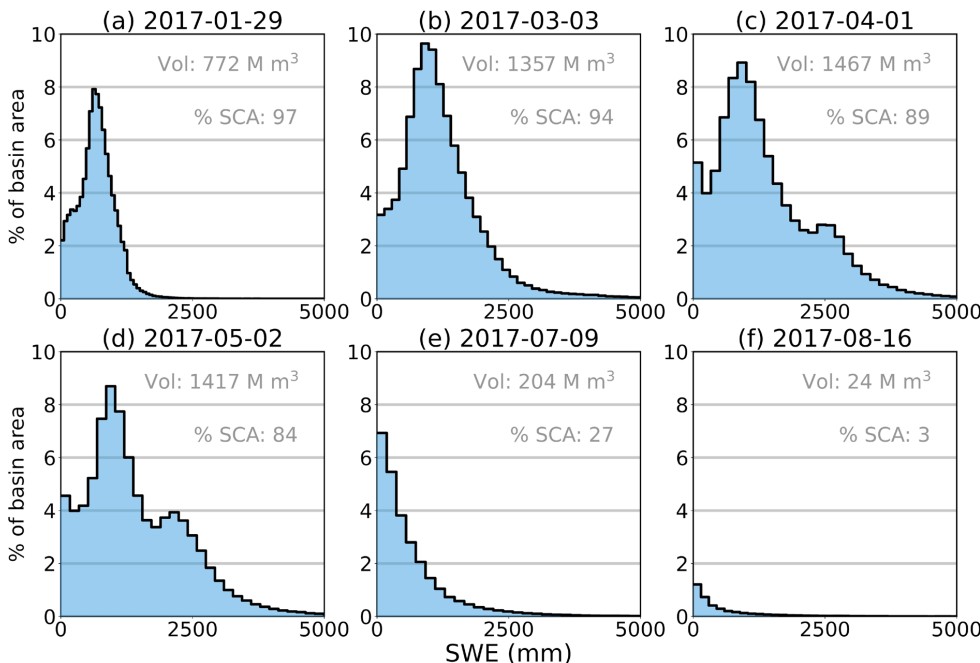

**Figure 6. Same as Figure 4 but for the record high WY 2017.**





### 3.3 Elevational Variations in Snow Accumulation

The first approach to characterizing the distribution of SWE across the basin consists of analyzing the elevational variations in snowpack storage and mean SWE. Albeit a simplistic approach, water managers and researchers use similar elevational information to understand where the snow is and when it will melt because precipitation generally increases with elevation and snow at higher elevations generally melts later in the season. Also, other processes such as avalanching can cause snow redistribution from higher to lower elevations, producing large snow deposits that can last late into the snowmelt season.

At the times around peak snow accumulation, SWE storage increases with elevation with maximum volumes stored in elevations between 2750-3250 m (45% of the basin area), and decreases with elevation above 3250 m (Figure 7 and Figure 8 only include elevations between 2250 m and 3500 m, although the general behavior outside this range is represented by the lower and upper elevation curves). Snowmelt rates are very different between lower and higher elevations. Lower elevations (e.g., 2250-2500 m) show early melt and snow disappearance while elevations above 3250 m display flatter SWE curves with delayed melting and slower melt rates, generally crossing lower elevation curves towards the end of the snow season, indicating later snow disappearance than at lower elevations. In WYs 2013-2015 (driest years), the lower elevation bands (e.g., 2250-2500 m) have lower snow storage and exhibit faster melt that starts earlier than at the higher elevations (e.g., 3250-3500 m). In WYs 2016 and 2017 (wettest snow years), the two elevation bands (2250-2500 m and 3250-3500 m) store almost the same SWE volume until the onset of melt at the lowest elevation band, after which the higher elevation band retains its snow storage through a period of two (2016) to three (2017) months before the onset of melt at higher elevations. This contrasting behavior highlights the very different dynamics in the mass and energy balances between the lower and higher elevations.



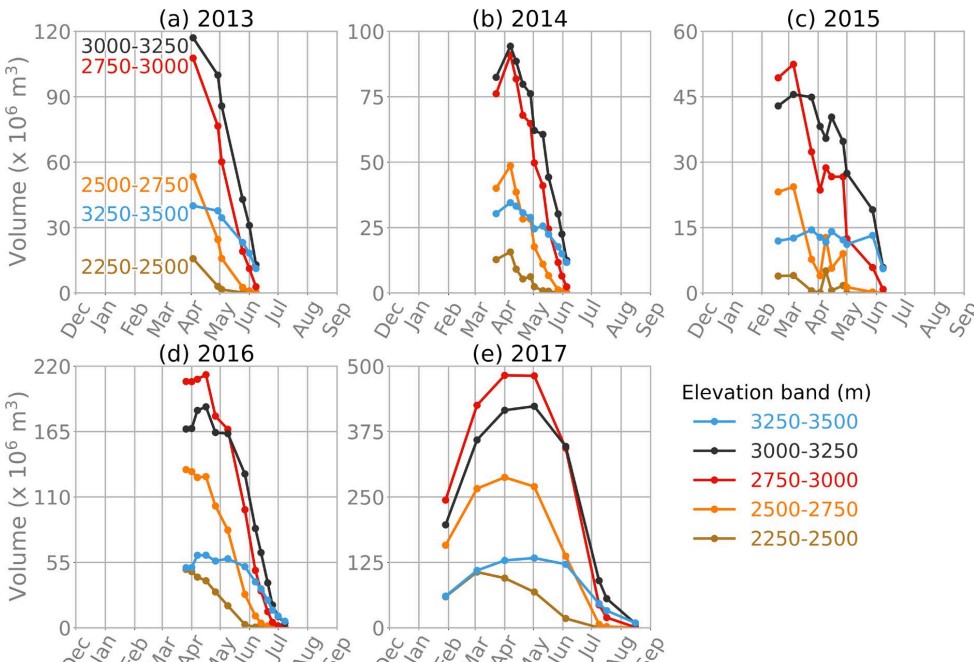

**Figure 7. SWE storage for 250-m elevation bands in the Tuolumne Basin for the study period WYs 2013-2017. The markers denote the flight dates for each year. Elevations are provided in meters.**

In contrast, mean SWE curves (Figure 8) show an increase in mean SWE with elevations to a maximum at elevations between 3000-3250 m (in black) at the time of peak, although slower melt rates lead to larger mean SWE at 3250-3500 m and above towards the end of the season. These differences between both SWE storage and mean SWE across elevations reflect a complex behavior in response to mass and energy balances across elevational gradients. The mean SWE curves (Figure 8) provide additional information to the SWE storage analysis (Figure 7) because they represent the mass per unit area, as opposed to the storage per elevation band, which is also a consequence of the basin morphology, in particular, variations in contributing areas across the vertical gradient (hypsometry in Figure 2c and d).



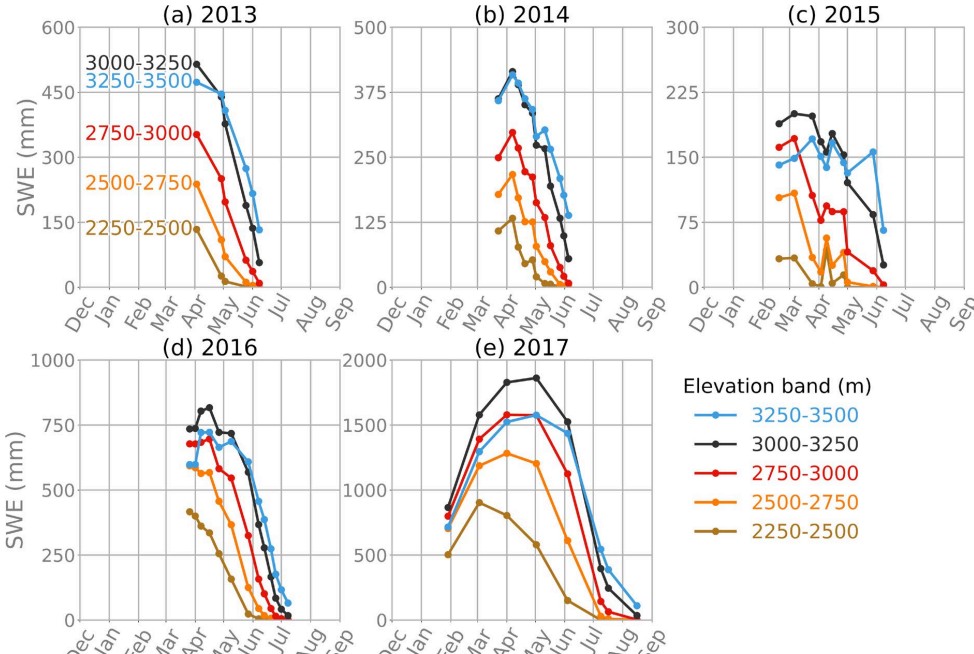

**Figure 8. Mean SWE for 250-m elevation bands in the Tuolumne Basin for the study period WYs 2013-2017. The markers denote the flight dates for each year. Elevations are provided in meters.**

Figure 7 and Figure 8 illustrate some of the variability of peak SWE across the basin, with later peak SWE dates with elevation (Figure S 1), with the exception of the extreme drought WY 2015. The intermittency of the snowpack during this extreme drought year makes the characterization of an individual peak difficult as the snow water equivalent curve displays intermittent accumulation and melt cycles. Also, in water years 2013 and 2016, the lowest elevation snow pillow peaked over a month before the first ASO acquisition, limiting the

applicability of using the ASO SWE to evaluate peak SWE in the lowest elevation bands for those years (e.g., < 2750 m in 2016). Peak dates can be delayed markedly across the elevational gradient of the basin, with snow accumulations at highest elevations peaking in late May - early June most years, while March - early April at mid and lower elevations (e.g., March 26 at 2500-2750m and May 9 above 3500 m in 2016, March 3 below 2500 m and May 2 and later above 3000 m in 2017).

**3.4    Spatially Resolved Date of Peak SWE Across the Basin**

Water years 2015, 2016 and 2017 are particularly useful for identifying spatial patterns of peak snow accumulation across the watershed, as they represent a transition from one of the driest years on record (WY 2015), to a near average year (WY 2016), and to one of the snowiest years on record (WY 2017). The ASO surveys for these years covered a long enough period to properly characterize the variation in the timing of peak

for each grid-cell across the watershed (10 flights between February 18 and June 8 in WY 2015, 13 flights between March 26 and July 8 in WY 2016, and 8 flights between January 29 and August 16 in WY 2017, Table 1). The following analyses are presented such that the focus is on the transition from the drought WY 2015 to



the extreme snow WY 2017. The snow conditions during WY 2013 and 2014 are used to fill in the gap between these extremes, and the results are discussed following the analysis of WYs 2015-2017 for completion.

Water year 2015 provides a unique opportunity to characterize one of the driest years on record (Figure 9). Generally, locations with shallow peak snow accumulations tend to have ephemeral snowpacks with multiple accumulation and melt cycles throughout the season (~ less than 20 cm of SWE at the point scale, (Trujillo and Molotch, 2014)). This intermittency makes the characterization of peak SWE challenging as individual precipitation events can deliver enough snow to create a local maxima in the SWE curve, which can occur even

later in the snow season (e.g., areas below 2500 m in Figure 7c and Figure 8c). This was the case in WY 2015, with late peaks (i.e., peaks later than basin peak SWE volume; green in Figure 9, emphasized in the elevation distributions on the lower left panel) occurring both at lower and higher elevations, but with intermediate elevations peaking at the same time as basin SWE storage (March 6, blue in in Figure 9). A precipitation event in early April delivered enough snow to cause local maxima that led to late peak dates at lower elevations, and

increased snow accumulations at higher elevations to a new peak, where the onset of spring melt was just being initiated (e.g., > 3250 m in Figure 7c and Figure 8c).

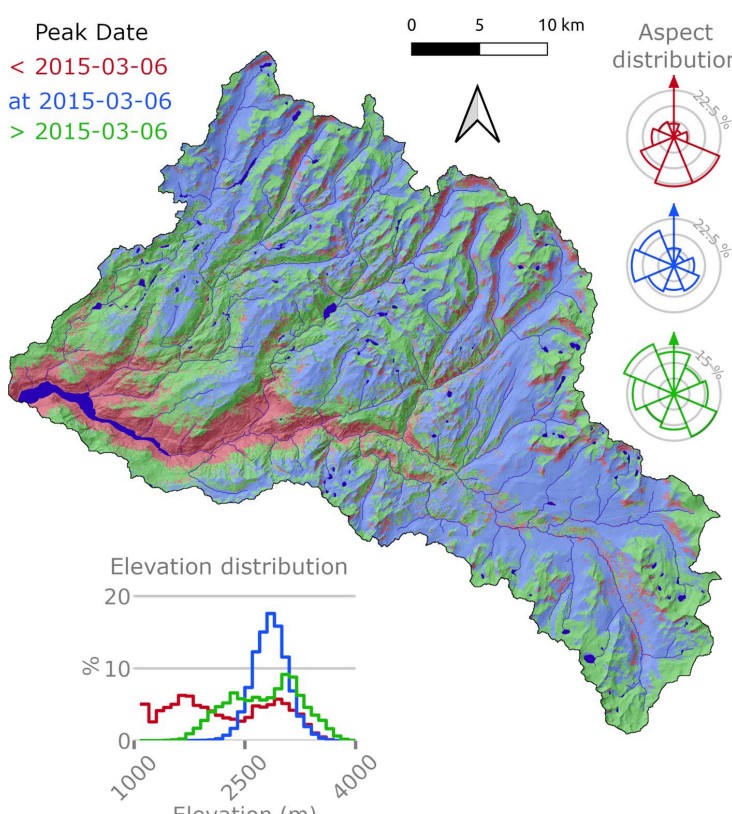

**Figure 9. Spatially resolved date of peak SWE across the Tuolumne basin for WY 2015. For simplicity,**
**the areas in which peak SWE is reached prior to the date of basin peak SWE storage (i.e., Figure 3a) is in red, at basin peak SWE storage in blue, and later than basin peak SWE storage in green. Aspect and**



**elevation distributions in these three categories are also shown following the same symbology, and frequencies are relative to the area in each category. "<" is used to indicate "earlier than", and ">" is used to indicate "later than".**


WY 2016 was characterized by a steady accumulation season with frequent precipitation events occurring throughout season (Figure 3b). This water year, the timing of peak SWE varied across the watershed with elevation and aspect partially explaining these variations (Figure 10). Elevation distributions indicate later peaks with elevation (lower left elevation distribution in Figure 10), while earlier peaks were generally aligned with

southerly aspects (upper right aspect distribution in Figure 10). Northerly aspects and higher elevations were associated with later peak SWE timing, and middle elevations and north-west/south-east aspects peaked at the same time as basin peak SWE storage (areas in blue in Figure 10). 65% of the basin area peaked prior to April 16, 29% peaked on April 16, and 6% peaked after April 16. Here, we must clarify that some of the aspect variations are not only caused by radiative effects, but also by snow redistribution by wind from the upwind

westerly slopes to the downwind easterly slopes at higher elevations. These effects are evident when closely evaluating the 3-m lidar snow depths, which provide an additional level of detail (not shown).

**Figure 10. Same as Figure 9 but for WY 2016.**




In contrast, the WY 2017 snow season was characterized by a steep rising limb of the SWE curve with the presence of multiple major precipitation events that delivered large amounts of snow in relatively short periods of time (Figure 3b), characteristic of large atmospheric river (AR) precipitation events (Lundquist et al., 2015; Neiman et al., 2008b, a; Guan et al., 2013, 2010). One large AR event occurred after the April 1 ASO flight (Figure 3b) and delivered considerable amounts of snow at the snow pillows located close to the north-eastern boundary of the watershed, all of which are located at elevations above 2800 m (stations HRS, SLI, VRG, DAN, GEM, and AGP). This event, however, did not deliver precipitation in the form of snow at the lower elevation stations (stations KIB, TUM, TNY, and WHW). In contrast with WY 2016, the timing of peak SWE in WY 2017 (Figure 11) was more evenly distributed around the timing of basin peak SWE storage (Figure 3a) with 39% and 30.5% of the area peaking before and after April 1, respectively, while the remaining 30.5% peaked on April 1. Once again, the elevation distributions (lower left panel, Figure 11) indicate later peaks with elevation, although the influence of elevation was less pronounced for the areas that peaked on April 1 and later. The influence of aspect on the timing of peak (upper right panel, Figure 11) is relatively similar to WY 2016 with south-westerly aspects peaking earlier, while north-west and south-east aspects were more frequent in the areas that peaked on April 1. However, the areas that peaked after April 1 were not as clearly aligned with northerly aspects as in WY 2016.

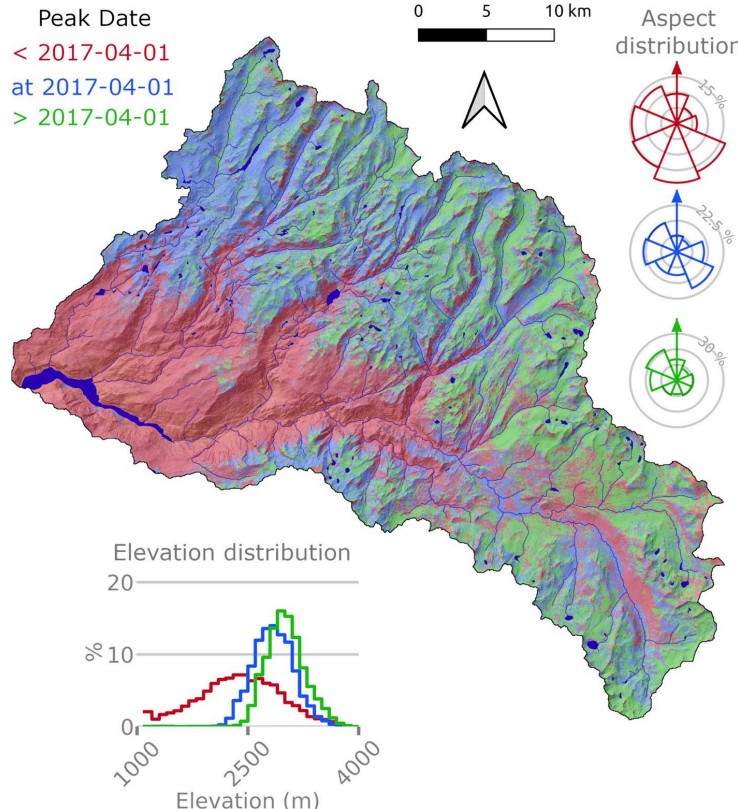



**Figure 11. Same as Figure 9 but for WY 2017.**

WYs 2013 and 2014 (Figure S 2 and Figure S 3) provide additional information for intermediate snow conditions. In WY 2013 (Figure S 2), most of the basin peaked on April 3 (82%), which was the time of the first ASO flight. As mentioned earlier, peak snow accumulation could have been reached just prior to the time of this

first flight at middle and lower elevations. However, snow pillow SWE indicates that the timing of the flight was adequate and most of the pillows peaked at or just prior to the flight (Figure 3). Similarly, later peak SWE was associated with higher elevations and northerly aspects and occurred over 17% of the basin. The same observation applies to WY 2014 (Figure S 3), although the distribution of peak around basin peak SWE was more evenly distributed with 19% peaking early, 52% peaking at the time of basin peak SWE (April 7), and

28% peaking after. The residual values in these percentages correspond to water bodies and areas without information.

### 3.5 Spatially Resolved Peak SWE

The analyses above illustrate the large range of variability in the timing of peak SWE across a large headwater catchment, with the earliest peaks reached as far as two months prior to the latest peaks, albeit this variability

greatly depends on each season's weather patterns. Here, we discuss the differences between the spatially distributed peak SWE maps and SWE at the timing of basin peak SWE storage, as described in the Data and Methods section.

During the drought year 2015, the smallest SWE differences (> 0 to 100 mm, red in Figure 12) are concentrated on the south-east facing slopes and middle elevations (2250 m – 3000 m). These differences increase with

elevation and towards the northern aspects, with the largest differences (> 300 mm, green in Figure 12) markedly aligned with the northerly slopes. The total volume of these SWE differences amounts to 42 M m$^3$ of water (23% negative bias relative to spatially resolved peak SWE – negative bias hereafter), which would correspond to the amount of SWE volume that would be unaccounted for if a single snowpack snapshot is obtained at the date of basin peak SWE volume (March 6 in 2015, Figure 3). Furthermore, if a single snowpack

snapshot is captured on April 3 (closest lidar survey date to the commonly used April 1), the total SWE volume unaccounted for would be 102 M m$^3$ of water (56% negative bias). The estimates and statistics for this year must be given some special considerations because of the challenges of characterizing a single peak value and date because of the ephemeral nature of snowpacks during a drought year, with multiple accumulation and melting events throughout the season.




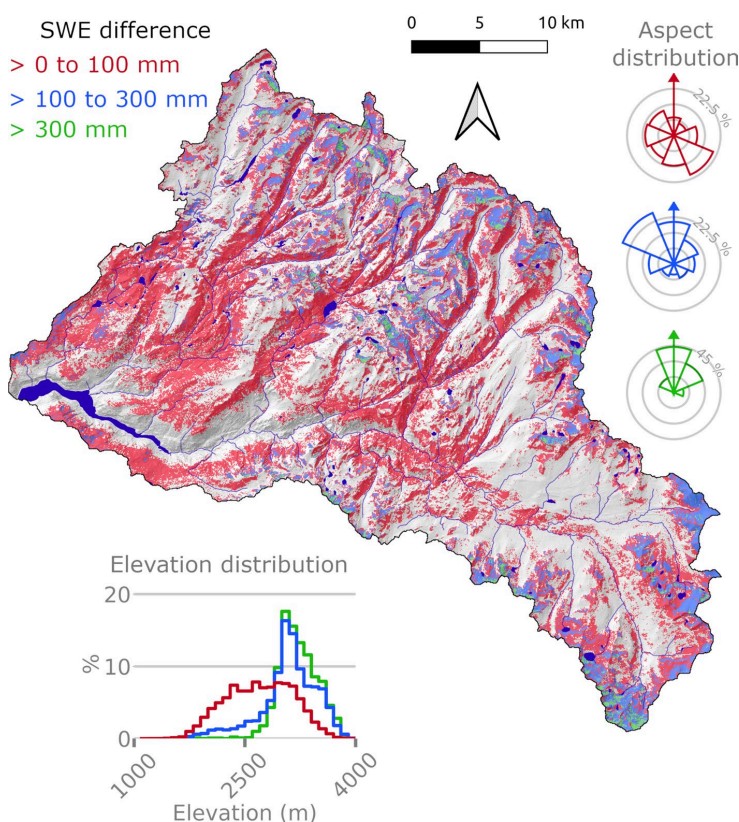

**Figure 12. Difference between the spatially resolved peak SWE minus SWE at basin peak SWE storage (i.e., Figure 3a) for WY 2015. The locations in white have a SWE difference of zero (e.g., areas that peak at the same time as basin SWE storage or areas without snow accumulation during the surveys). ">" is used to indicate "greater than". The area in red is 38 % of the basin area, in blue is 9.1 %, and in green is 1.3 %.**

In the average WY 2016 (Figure 13) SWE differences tend to decrease with elevation, contrary to what was observed in 2015. Differences of less than 100 mm are concentrated at the highest elevations, but the aspect distribution is similar to that of the entire watershed (Figure 2a and b). Differences between 100 mm and 300 mm are present in all aspects but predominantly along the southeastern slopes. Lastly, the largest differences (> 300 mm) are observed predominantly on northerly slopes, although they cover a very small area (1.3 % of basin area). The total volume of these SWE differences amounts to 62 M m$^3$ of water (9 % negative bias), which would correspond to the amount of SWE volume that would be unaccounted for if a single snowpack snapshot is obtained at the date of basin peak SWE volume (April 16 in 2016, Figure 3). If a single snowpack snapshot is captured on April 1, the total SWE volume unaccounted for would be 82 M m$^3$ of water (12 % negative bias).





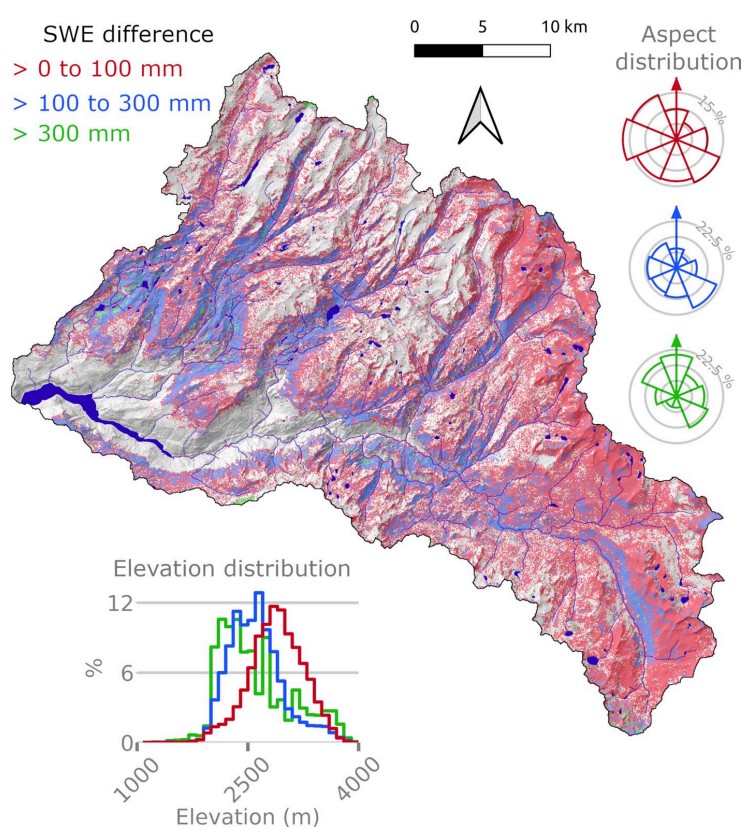

**Figure 13. Same as Figure 12 but for WY 2016. The area in red is 40.6 % of the basin area, in blue is 17.7**
**% , and in green is 1.3 %.**

In the snowiest WY 2017 (Figure 14), SWE differences have different and complex characteristics when compared to the driest and average years prior. The largest differences (> 300 mm, in green) are predominantly present both at low and high elevations, with a bimodal distribution with a low at the middle elevations around 2500 m. The lowest differences (> 0 to 100 mm, in red) are concentrated between 2500-3000 m, although with a mild second mode at elevations around 1500 m. Similarly, differences between 100 and 300 mm are concentrated around the same elevations between 2500-3000 m, but the distribution does not have a second mode as the other two cases. When analyzed in parallel with the date of peak SWE (Figure 11), the lower elevation modes on these distributions are within the range that peaks earlier, while the higher elevation modes are located in areas that peak later than the timing of basin peak volume (April 1 in 2017). This indicates that early precipitation events delivered large amounts of snow at lower elevations, but melt was well underway by the time of peak basin snow volume (April 1). At the higher elevations, a second SWE peak occurred in 2017 due to multiple late season precipitation events after April 1. Aspect distributions in 2017 do not show much variation between the three SWE difference ranges, with a signal that indicates little influence from aspect during this extreme snowiest year. The total volume of these SWE differences amounts to 149 M m$^3$ of water (9



% negative bias), which would correspond to the amount of SWE volume that would be unaccounted for if a single snowpack snapshot is obtained at the date of basin peak SWE volume (April 1 in 2017, Figure 3).

Lastly, WYs 2013 and 2014 (Figure S 4 and Figure S 5, respectively) show similar distribution patterns of peak SWE differences, with the areas with SWE differences greater than zero concentrated in the middle elevations

495     (around 3000 m) and northerly and north-westerly aspects. However, the differences in WY 2014 increase with elevation and towards northerly aspects. This is partly due to the lack of early season flights in WY 2013, which impeded a better determination of the peak in areas that may have peaked earlier than the first flight. The total volumes of these SWE differences amount to 28 M m$^3$ (7 % negative bias) and 27 M m$^3$ (8 % negative bias) of water in WYs 2013 and 2014, respectively, which would correspond to the amount of SWE volume that would

500     be unaccounted for if a single snowpack snapshot is obtained at the date of basin peak SWE volume (April 3 in 2013, and April 7 in 2014, Figure 3).

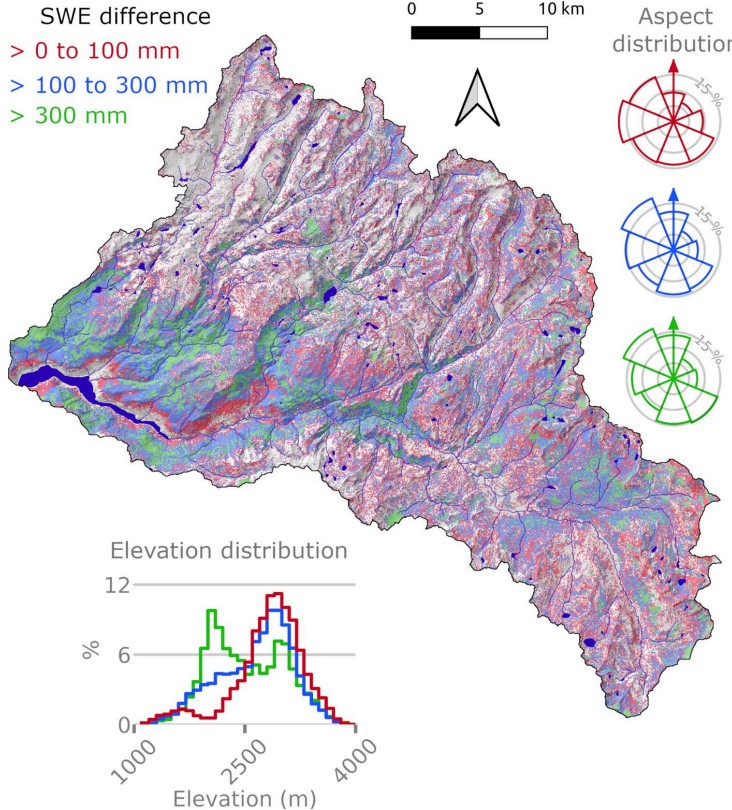

**Figure 14. Same as Figure 12 but for WY 2017. The area in red is 26.4 % of the basin area, in blue is 28.8**

505     **%, and in green is 11.8 %.**



## 4    Discussion

In this study, we have taken advantage of the ranging snowpack conditions during the period between WYs 2013 and 2017 to characterize the spatial and temporal features of snow distribution (SWE) in a snow dominated headwater catchment of 1180 km². There are many analyses of snow distribution from lidar and Structure from Motion (SfM) snow depths at similar or higher resolutions in the western US (Trujillo et al., 2009, 2007; Trujillo and Lehning, 2015; e.g., Deems et al., 2006, 2008; Kirchner et al., 2014; Pflug and Lundquist, 2020; Meyer and Skiles, 2019), the Canadian Rockies (e.g., Pelto et al., 2019), the Alps (Grünewald et al., 2010, 2014; Schirmer et al., 2011; Schirmer and Lehning, 2011; Mott et al., 2011; e.g., Dadic et al., 2010), the Andes (e.g., Mendoza et al., 2020; Shaw et al., 2020), and polar environments (Williams et al., 2013; Trujillo et al., 2016; Sommer et al., 2018; e.g., Petrich et al., 2012), among many others. However, no similar analyses have been performed with snow water equivalent and the combined spatial and temporal coverage herein. These aforementioned studies were focused on individual or a small number of snapshots of the snow distribution, and over smaller domains, in great part limited by the available datasets and the contemporary state of development of snow measurement technologies.

There are many advantages in analyzing SWE instead of snow depth for hydrological purposes and water resources applications. SWE provides information about the water storage in headwater catchments, which cannot be inferred from snow depth alone; a spatial estimate of mean snow density is also required. Distributed SWE is necessary for examining how snowpack accumulation and melt varies throughout a watershed. Similarly, decreases in snow depth do not equate to snowmelt because bulk snow density increases translate into snow depth decreases without melt necessarily. The snowpack densification process is dynamic and ubiquitous throughout the snow season (Kojima, 1967, 1975; Bormann et al., 2013; Sturm and Holmgren, 1998; Mizukami and Perica, 2008). This can be easily appreciated when observing co-located time series of snow depth and snow density (e.g., snow pillows and ultrasonic snow depth sensors at SNOTEL stations), in which seasonal peak snow depth is generally reached at the same time or prior to peak SWE. Often, peak SWE levels can be maintained for an extended period of time until melting is initiated, as long as atmospheric losses (e.g. evaporation and sublimation) are minimal, while at the same time snow depth experiences a decreasing limb because of densification. In consequence, analyzing distributed lidar time series of snow depth (e.g., airborne lidar, TLS, SfM, or UAS based lidar) as a proxy for snowpack storage would not allow for an accurate assessment of the progression in snowpack storage, peak snow accumulation (SWE) and timing, and differential melt rates, all of which are features of snowpacks in headwater catchments. Here, although comprehensive, our analyses were limited to these spatiotemporal features of snowpack storage over the Tuolumne basin.

The study period notably included the transition between one of the driest (WY 2015) to one of the wettest (WY 2017) snow seasons in recorded history in the Sierra Nevada. Peak snowpack storage in the Tuolumne basin above the Hetch Hetchy reservoir ranged from 142 M m³ to 1467 M m³. Such a wide range in snowpack storage highlights hydroclimatic vulnerability in regions like California that depend on winter snowpacks as the principal water resource. The effects of such extremes in the region have been well documented from the dust bowl in the 1930s (Schubert et al., 2004; Hornbeck, 2012) to the extreme snow year 2017, during which infrastructure failure, such as the spillway damage in the Oroville Dam (Henn et al., 2020), threatened downstream communities.



Generally, SWE empirical distributions were unimodal and truncated at zero around peak accumulation, and as the melting seasons progressed, they showed a decaying function (Figure 4-Figure 6). Only in some years with late season precipitation events delivering snow at higher elevations (e.g., WY 2017), these distributions showed a second mode towards higher SWE values. These empirical distributions are similar to those of lidar
snow depths in the Tuolumne at the same basin scale (Mason et al., 2019; Mason, 2020), and other mountain environments in smaller domains and at higher resolutions (< 50 m) (Mendoza et al., 2020).

The elevational variations of both snow volume storage (Figure 7) and mean SWE (Figure 8) reflect the dynamics of accumulation and melt processes and the influence of basin morphology on snow storage. The general elevational increase in snow storage to a maximum at elevations between 2750-3250 m is a combination
of the increase in SWE with elevation and the greater apportionment of basin area in those middle elevations. The increase in snow accumulations with elevation at distributed scales has been well established using lidar snow depths in mountain environments (Lehning et al., 2011; Grünewald et al., 2014; Kirchner et al., 2014; Trujillo et al., 2009). These studies have identified that snow accumulation (i.e., snow depth) increases with elevation to a maximum above which there is an observed decrease in accumulations. This decrease could be
caused by cloud water depletion (Kirchner et al., 2014), and increases in snow redistribution by wind, sublimation and direct sun exposure above the tree-line (Grünewald et al., 2014; Trujillo et al., 2009). Other snow processes such as avalanching can also redistribute snow from higher to lower elevations affecting the snow distribution at these elevations (Mock and Birkeland, 2020). The elevational variations in snow volume and mean SWE at the Tuolumne Basin show similar maxima at the middle elevations and a subsequent
decrease, although the maxima occur at different elevations for mean SWE because hypsometry is ignored by definition.

Such elevational variations in water storage have implications for both the timing of the largest snowmelt pulse and the travel time of snowmelt inputs to the outlet of the basin. In the event of rain-on-snow (ROS) events, the storage in these middle elevations at the time of the event can be an important factor in resulting flows at the
outlet. Luo and Harlin (2003) explored the relationship between basin hypsometry and hydrologic response using a theoretical travel time distribution approach, building on the previous work relating hydrologic response to basin morphology (Rodríguez-Iturbe et al., 1979; Gupta et al., 1980; Howard, 1990). In their approach, Luo and Harling (2003) relate the hypsometric curve (e.g., Figure 2d) to basin hydrologic response because the watershed geomorphology determines the distribution of potential energy. The potential energy can then be
described by the hypsometry attributes when the hypsometric curve is treated as a cumulative probability distribution. Vivoni et al. (2008) later showed that basin hypsometry exerts control on surface and subsurface runoff partitioning. These works generally explore the hydrologic response to rainfall, and few examples exist of similar analyses in snow dominated watersheds. The results here further highlight the added complexity of runoff processes in snow dominated watersheds, with the hypsometry of the basin also partially controlling
basin snow storage and snowmelt timing because of the relationships between elevation, snowfall and snowpack energetics.

One of the main differences in the hydrology of snow dominated versus rainfall dominated watersheds lies in the process of differential snowmelt that dominates headwater catchments in the Sierra Nevada and other regions. This differential melting consists of partial surface water inputs from areas where snowmelt starts and
stops as the season progresses and the snowpack depletes according to the local water and energy balances,



resulting in an elongated streamflow pulse over several weeks when compared to that caused by rainfall. In the Tuolumne Basin, this differential melting is evident in the elevational variations in snow volume (Figure 7) and mean SWE (Figure 8) as the season progresses, with slower and later melt at higher elevations. Few examples exist of analyses of these differential melting processes at distributed watershed scales and their relationship to

hydrologic response.

With the expansion of integrated lidar/modeling datasets like the one in this study, with full spatial coverage across a watershed and throughout the melt season, new analyses of the relationships between geomorphologic basin characteristics (e.g., hypsometry, river network structure, slope and aspects), basin snow storage, and hydrologic response will be possible for snow dominated basins. Some previous analysis based on snow and

hydrological modeling include Comola et al. (2015), in which an artificial rotation of a 43 km$^2$ snow dominated headwater catchment in the Swiss Alps was used to explore the effects of radiative fluxes and basin orientation on the hydrological response, based on results from the Alpine3D and Streamflow models. They concluded that the effects of these rotations, simulating changing shortwave inputs, became muted with increasing contributing area because aspect distributions stabilized regardless of basin orientation. Extending these types of analyses of

the effects of basin morphology on snow storage, snow melt and the corresponding hydrologic response will become possible with accurate distributed snow measurements and improved models using data assimilation of distributed information (e.g., Hedrick et al., 2018b, 2020).

The spatially resolved date of peak SWE (Figure 9-Figure 11 and Figure S 2-Figure S 3) explicitly demonstrates the large variability in the timing of peak SWE across a watershed of the size of the Tuolumne basin (1180

km$^2$). Accordingly, the assumption that peak SWE is reached at similar times across a watershed of this size should be carefully considered, given that the controlling processes in the mass and energy balances are highly heterogeneous in both time and space. This observation is also important to illustrate that the onset of melt, which is generally well represented by the timing of peak SWE (Trujillo and Molotch, 2014), is highly variable across headwater catchments with strong elevational gradients and clear aspects variations (Figure 2).

Additionally, there are relevant implications about what the optimal timing of snow surveying should be to appropriately account for snow water storage in headwater catchments, both from in-situ measurements (e.g., snow courses and snow pillow stations) and remote sensing (e.g., airborne lidar or satellite sensors). This topic is particularly pertinent as federal, state, and local agencies (e.g., California Department of Water Resources (CDWR), NASA, Denver Water and many others) explore new and extended deployments of airborne and

satellite snow measurement programs. These variations in the timing of peak and the onset of melt also have important implications for water managers and reservoir operators, as the variable timing of snow melt inputs will strongly affect decision making regarding when to hold, release and use water for energy generation.

As the overarching goal of global snow monitoring gains traction thanks to the recent improvements in snow remote sensing and modeling, discussions focus on the optimal timing and frequency of satellite missions and

airborne surveys (e.g., Hall and Vuyovich, 2020). Though it is now well-understood that spatial patterns of snow distribution are relatively consistent inter- and intra-seasonally (Sturm and Wagner, 2010; Deems et al., 2008; Pflug and Lundquist, 2020; Schirmer et al., 2011; Schirmer and Lehning, 2011), this work shows that a single survey is inadequate for capturing the evolving nature of SWE storage throughout the year. Figures 9-11 show that if one survey was used near the date of peak SWE, a significant portion of water that has already

melted or has yet to accumulate would not be accounted for due to hypsometric and climatologic factors.



Though this study only concerns snapshots of the ever-evolving SWE distribution, we can see that at the very least, survey dates before, during, and after the date of peak SWE better constrain accumulation and melt process in this large basin. This approach is already being taken up by the California Aerial Remote Sensing of Snow (ARSS) program, which now requests three to five surveys at a minimum from ASO, Inc. for all of the

basins throughout the Sierra Nevada.

The spatially resolved date of peak SWE shows the wide range of variability in the timing of peak SWE across a watershed of this size (i.e., 1180 km$^2$) across very different snow seasons. Interannually, snow accumulations in dry years can peak much earlier when compared to average or above average snow years (e.g., Figure 3), particularly in regions with the climatic variability of the Sierra Nevada region, even though the April 1 date of

basin peak SWE volume appears to be representative on the average. Intra-seasonally, the date of peak SWE across a watershed can also vary by several weeks (e.g., Figure 9 versus Figure 11), with precipitation events and differential melting affecting the timing of peak SWE. We note here that the time intervals between flights add an inherent uncertainty to the dates of peak SWE of the order of half the interval (e.g., ± a few days or one week).

Consequently, these results bring up the following question: What are the limitations of obtaining a single remote sensing survey of the snowpack at or around the time of maximum basin snow storage? It is clear from this study that a single survey does not adequately capture the evolving nature of SWE storage throughout the year. The spatially resolved peak SWE analysis (Figure 12-Figure 14, and Figure S 4 and Figure S 5) shows that a single survey around peak SWE, although representative of peak basin snow volume, would lead to an

underestimation of spatially resolved peak SWE across the watershed. This underestimation was between 27 M m$^3$ and 149 M m$^3$, corresponding to negative biases between 7 % and 12 % for about average to extremely snowy water years (2013-2014, 2016-2017). For the special case of the extreme drought year 2015, which exhibited multiple snow accumulation and melt cycles throughout the snow season, this underestimation ranged widely between 42 M m$^3$ and 102 M m$^3$ (23 % and 56 % negative bias), depending on the lidar survey date.

However, the interpretation of these results during drought years with multiple accumulation and melt cycles should be given special consideration since the snowpack has multiple local SWE maxima throughout the season.

This underestimated volume can be thought of as the portion of water that has already melted or has yet to accumulate at the time of peak basin storage (e.g., April 1). A single peak SWE survey would limit the

characterization of snow distribution patterns that can be used to inform snow, atmospheric and hydrological models (e.g., snowpacks at lower elevations that melt out prior to an April survey). Though this study only concerns snapshots of the temporally dynamic SWE distribution, it reveals that survey dates before, during, and after the date of peak SWE better constrain accumulation and melt process in this large basin. In fact, the California Aerial Remote Sensing of Snow (ARSS) program at the Department of Water Resources (DWR) is

now operationally implementing this approach by annually requesting three to five surveys at a minimum from ASO, Inc. for all of the basins throughout the Sierra Nevada. Lastly, as with any technology, there are tradeoffs; more frequent airborne surveys can be associated with higher costs, or in the case of satellite missions, higher spatial resolutions can be associated with lower temporal resolutions. Our results provide insights into some of these tradeoffs when obtaining distributed measurements in headwater catchments.



**5    Conclusions**

In this study, we present a comprehensive analysis of the spatial and temporal dynamics of snow distribution in a headwater catchment in the Sierra Nevada, California. Located above Hetch Hetchy Reservoir, the Tuolumne River Basin contains a contributing area of 1180 km² and an elevational range from 1150 m to 4000 m. The dataset used is the product of ASO lidar snow depths and iSnobal modeled snow densities at 50-m resolution for
the water years 2013-2017. A direct insertion approach is used in the model to constrain the snow depth distribution to the lidar observations.

The integrated SWE dataset analyzed in this study, where the snow depth component of SWE is periodically constrained by aerial lidar, is an alternative to statistical approaches that relate SWE to snow depth from historical station data or combine interpolated snow depths and snow densities across a landscape. With the
continuing pursuit of improvements in remote sensing of snow properties in forested and mountainous terrain (e.g., NASA SnowEx (https://snow.nasa.gov/campaigns/snowex) and beyond), this study describes how high-resolution snow models, when combined with periodic remote sensing products, can be used to analyze distributed SWE information over larger domains and mountain ranges, and throughout both accumulation and ablation seasons.

The study period includes the progression from one of the driest to one of the snowiest years on record in the California Sierra Nevada, illustrating the wide range in snowpack water storage and snowpack dynamics that the region experiences. The analysis demonstrates how the distribution of SWE in a headwater catchment evolves throughout the season, with empirical probability distributions over the catchments varying from unimodal and bimodal distributions to decaying distributions towards the end of the snow seasons. Snowpack water storage is
greatest at the middle elevations (2750-3250 m) because of the combined increases in snow accumulation and larger proportions of the basin area concentrated at these elevations. Lower, middle, and higher elevations contribute to streamflow at different times in the season, as spring melt is initiated at later times with increasing elevation and snowmelt rates vary largely across the gradient. The date of peak SWE also varies largely across the basin and between years, illustrating that a single point-in-time definition of peak SWE timing (e.g., April 1)
does not capture the temporal dynamics over a headwater catchment of this size. It follows from this observation that a single distributed snow survey at or around the date of basin peak SWE volume would likewise not capture peak snowpack conditions across the watershed and would lead to an underestimation of the amount of snowmelt water that would contribute to streamflow and the soil and aquifer system in a given year. For WYs 2013-2017, this underestimation for the Tuolumne Basin ranges between 27 and 149 M m³, corresponding to
negative biases between 7 % and 12 % of peak snowpack storage.

Results from this study provide insights into the type of data needed to advance our current understanding of snowpack dynamics in regions where snowpack water resources are critical. Quantifying the effects of the large meteorological variability in the Sierra Nevada region on water resources requires frequent and accurate snowpack estimates, with spatial and temporal resolutions capable of resolving the natural variability in
complex terrain. The need for these types of datasets and the novel research that they provide is becoming increasingly evident, as these critical regions are exposed to frequent snow droughts, floods and large forest fires, among other disturbances, that have major impacts on ecosystems and societies. This analysis improves reservoir operations through understanding where the snow is and when it melts, supports decision making in the experimental design and operational deployment of airborne and satellite missions for snow mapping, and



provides opportunities to enhance ground survey design and planning for water supply forecasting. Similar

analyses across different regions will be particularly relevant to support future global snow mapping efforts.



**Code/Data availability**

The full Dataset as analyzed in this study is published in the USDA Ag Data Commons data repository

(https://doi.org/10.15482/USDA.ADC/29403614).

**Author contribution**

ET designed and performed the analyses and carried them out, with contributions from AH. ET and AH prepared the manuscript with contributions from DM.

**Competing interest**

The authors declare that they have no conflict of interest.

**Acknowledgements**

This research was supported by the USDA-ARS CRIS Project, Ecohydrology of Mountainous Terrain in a Changing Climate (2052-13610-012-00D), the USDA-ARS cooperative agreement with UC-Merced, Sierra Nevada Water Supply Forecasting Project (58-2052-7-002), and the USDA-ARS cooperative agreement with

Boise State University, Snow Water Supply Forecasting - Streamflow Modeling (59-2052-0-001). Any reference to specific equipment types or manufacturers is for information purposes and does not represent product endorsement or recommendation. USDA is an equal opportunity provider and employer. The dataset used in this study was generated using the reproducible framework in Hedrick et al. (2018b, https://zenodo.org/record/1343647, 2020).




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





## Supplementary Information

**Figure S 1. Dates of peak SWE per elevation bands across the full basin elevational gradient.**

**Figure S 2. Spatial organization of peak SWE across the Tuolumne basin for WY 2013. For simplicity, the locations in which peak SWE is reached prior to the date of basin peak SWE storage (i.e., Figure 3a) are in red, at basin peak SWE storage in blue, and later than basin peak SWE storage in green. Aspect**
**and elevation distributions in these three categories are also shown following the same symbology. For WY 2013, early peaks were not detected because the lidar flights did not covered a period prior to 2013-04-03. "<" is used to indicate "earlier than", and ">" is used to indicate "later than".**

**Figure S 3. Same as Figure S 2 and Figure 9-Figure 11 but for WY 2014.**

**Figure S 4. Difference between the spatially resolved peak SWE minus SWE at basin peak SWE storage (i.e., Figure 3a) at the Tuolumne Basin for WY 2013. The locations in white have a SWE difference of zero (i.e., areas that peak at the same time as basin SWE storage or did not accumulate any snow). ">" is used to indicate "greater than". The area in red is 10.3 % of the basin area, in blue is 5.2 %, and in green**
**is 1.9 %.**

**Figure S 5. Same as Figure S 4 and Figure 12-Figure 14 but for WY 2014. The area in red is 26.7 % of the basin area, in blue is 5.1 %, and in green is 0.9 %.**