# Peer review of "Spatial and temporal features of snow water equivalent across a headwater catchment in the Sierra Nevada, USA"

_EGUsphere, 2025_

## Referee Comment (RC1)

**Review of: SPATIAL AND TEMPORAL FEATURES OF SNOW WATER EQUIVALENT ACROSS A HEADWATER CATCHMENT IN THE SIERRA NEVADA**
**Trujillo et al, HESS**

**Summary**

In this study, the authors analyze the Airborne Snow Observatory (ASO) data in the Tuolumne River Basin in California. This data comprises 50-m SWE derived from airborne lidar-based snow depth, multiplied by a snow density estimate simulated using iSNOBAL. The data spans 5 years, with a variety of interannual snow conditions and several acquisitions per year. The authors assess histograms of the spatial distribution of SWE in each acquisition, elevation-dependence of total SWE volume and mean SWE, the pixel-wise estimated difference in peak SWE timing from the basin-wide peak SWE timing, and the error in SWE if basin-wide peak SWE timing is assumed to represent peak SWE timing at all pixels. In my view, the most interesting results are those that show that the spatial patterns of potential peak SWE timing and quantity errors vary interannually. There is some discussion of why these patterns are observed. However, I think the study needs a substantially clarified description of its motivations to show why the findings are novel and applicable beyond the specific time and space domain of the study, and to illustrate a clear purpose to the analyses. There are also several unsubstantiated and/or under-quantified claims and under-investigated assumptions. These are detailed in the major and minor comments below. I think the study may be suitable for publication following substantial revisions.

**Major comments**

*Motivation:* I think the introduction should be revised thoroughly to clarify why the specific analyses conducted are valuable. As is, I finished reading the introduction with only a glimmer of understanding about the value of the work that is described in the rest of the manuscript. I think the motivation needs to be stated more clearly in the introduction, and some of the analyses could be improved to clarify and enhance the contributions of the paper. For example, Figures 4-6 could be enhanced by reporting kurtosis and skewness, or perhaps by comparing them to the existing literature on snow depletion curves. This could tie the results better to modeling efforts. Figures 9-14 have some interesting results (essentially that the spatial patterns of pixel-wise error in peak SWE timing or magnitude vary interannually), but the introduction could better set the reader up for these results by acknowledging existing work on spatial variability of peak SWE timing, and describing the importance of the specific contributions here (fine-scale observations; SWE rather than depth; interannual variability of spatial patterns).

*Peak SWE and date of peak SWE:* The peak SWE and date of peak SWE calculations both depend strongly on when lidar acquisitions occurred. Figure 3 and the accompanying text seem designed to address this issue, but they are limited in important ways. I'm not convinced that snow pillow peak SWE is representative of when basin peak SWE would be expected to occur. We know that snow pillows are somewhat biased relative to their immediately surrounding terrain (Herbert et al., 2024), and I suspect are even more biased with respect to timing, due to canopy gap effects. I'd like to see evidence of whether they can be relied upon to represent basin peak SWE timing more broadly. Koshkin and Marshall (2025) – also a preprint – face the same issue of requiring a date of basin-wide peak SWE in analyses based on ASO data. They use a modeled data product to estimate the date of basin-wide peak SWE. While this approach relies on the model being approximately correct, it at least has a different set of strengths and limitations and could be valuable to reproduce here using the iSnobal simulations. I think the text in lines 269-273 is much too cavalier about this issue, and it should also be addressed in

the methods – readers will be wondering how dates of peak SWE are obtained given the intermittent timing of lidar acquisitions. It would also be helpful to at the very least quantify the variability in peak SWE timing among SNOTEL stations, and compare this to the difference between a SNOTEL- and ASO-based date of peak SWE. You might also consider rewording the analysis to avoid claiming that the dates you identify represent a true date of peak SWE, and instead acknowledge that these are defined only within the domain of the available lidar acquisitions. Overall, there are a number of approaches that could be taken to deal with this issue, but I think assigning the date of peak SWE based on dates of lidar acquisition (at the pixel or basin scale) is problematic and needs to be either justified more extensively (with uncertainty quantified) or avoided.

It's also worth noting that the fact that dates of peak SWE vary spatially is well-established using modeled and observational datasets (most relevant is Montoya et al., 2014; figure 4 in Marshall et al., 2019 also provides relevant results, although on a larger spatial scale). There may also be other analyses of this that I'm less familiar with. Figure 9 is important in this aspect: One of the most interesting things about this figure is the earlier peak SWE date on southeast-facing, high-elevation slopes. What's happening there? Avalanching? I think this figure highlights a potential missed opportunity: there are some spatial patterns of interest here that couldn't be discerned with either in situ observations or larger-scale models, as has been done previously. I also think the most interesting parts of this figure could be better highlighted if claims of peak SWE were avoided, and the analysis was presented relative to the lidar acquisition with maximum basin-wide SWE. Similarly, the interannual variability of the pixel-wide peak SWE timing relative to basin-wide peak SWE timing is a more novel contribution, and this finding and the reasons for it could be more effectively highlighted. To what extent is this driven by the interannual climate characteristics highlighted in the text, vs the chance of individual storm events (e.g. a late-season storm that falls as snow at high elevations only)?

*Lidar snow depth errors:* I'd like to see a slightly more thorough approach to describing the errors in the lidar data, which I don't think are fully known. The abstract mentions a <20cm vertical accuracy without citation, while the methods (line 180) cite a 0.08 m uncertainty for the 50-m depth product based on Painter et al. (2016). Painter et al. (2016) actually claim a 0.02 m uncertainty for the 50-m product, and a 0.08 m uncertainty for the 3-m product, though the logic of the uncertainty scaling is not fully explained there. As far as I understand, these uncertainties are based on depth surveys in relatively flat areas in Tuolumne Meadows, and therefore represent best-case estimates, given the potential for uncertainties in the beam angle to propagate in complex terrain. Painter et al. (2016) note: "field validation and geometric analyses for complex terrain are needed going forward to have a more comprehensive understanding of the ASO snow depth uncertainties." I haven't seen publications that develop this comprehensive understanding and I'm not arguing that the authors need to do that here. However, because the present study is predicated almost entirely on accurate ASO data, the text describing the dataset would benefit from more acknowledgement of the observational uncertainty – which is currently done thoroughly for the modeled density, but not for depth. I'm listing this as a major comment because I think it's important, but it could realistically probably be accomplished with a few minor text edits or additional sentences that refer to the broader literature on airborne lidar uncertainty.

**Minor comments**

Line 9: Does the <20cm vertical accuracy cited here refer to snow depth or SWE? The sentence just says "snowpack."

Line 11: Suggest "from lidar-derived snow depths"

Line 15: It's hard to see what this "widest range in recorded history" could be compared to, given that we don't have multi-year lidar scanning for most basins. Please clarify within the sentence.

Line 20: It's hard to understand what "partially explained by elevation and aspect" is doing in this sentence.

Line 22: Is there also a model evaluation application?

Lines 30-31: Could you provide a range for the regional estimates, as you did for the west-wide estimate?

Line 45: Is this really stochastic? Or maybe variable?

Line 51-65: This is a style suggestion, not a requirement, but I think this section would be easier to follow if it was edited to focus on the findings with the in-text citation at the end of each sentence, rather than stating "X authors found y." As written, it puts the emphasis on the authors and study, rather than on the finding. At the end of this paragraph, I'm not sure what I'm supposed to take away from it: April 1 SWE is problematic but useful? What is the evidence that it remains useful, despite the problems presented? I don't doubt the utility of April 1 SWE but I think it's not presented fully – could you provide a citation showing predictive power of seasonal streamflow based on April 1 SWE, perhaps?

Line 126: I suggest removing "natural" – for example, I don't think we know for sure that the 2015 snow drought isn't attributable to anthropogenic climate change.

Figure 2: (c) and (d) provide the same information; you could consider combing the two. Is the aspect coloring scheme in a and b reasonably colorblind-friendly?

Line 213: 2017 was a big snow year, but I don't think either of the cited papers supports it being one of the biggest years on record, and Figure 2 in Marshall et al. (2024) suggests that "one of the biggest years on record" is an overstatement. I checked the April 1 SWE data for the Tuolumne snow course, and 2017 appears to be the fourth-largest year from 1930-2024. This issue comes up throughout the manuscript, and should be fixed throughout.

[Figure]

**Figure 1.** April 1 SWE at the Tuolumne snow course.

Line 230: Unless these were selected to be statistically representative, avoid the word "representative" or detail how these were selected.

Line 233: Could you provide names for these two different variables to avoid confusion?

Line 256: If "peak snow accumulations" and "peak SWE" are used interchangeably, why not just use "peak SWE"? "Peak snow accumulations" risks implying a reference to seasonal-total maximum snow accumulation.

Line 257: Peak SWE, not peak snow, right? "Snow" is ambiguous as depth vs SWE.

Line 259-262: I can't understand the purpose of these sentences in this location.

Line 266-267: Need to provide evidence that 2017 had record snow accumulation, or avoid claiming it.

Line 285: "Basin SWE" implies that this refers to is the distribution of basin-total SWE across samples – but this section is really referring to the within-basin spatial distribution. The emphasis in this sentence on the ASO and ARS snow obs/modeling programs seems overly adulatory for the results section, and might be more appropriate in the introduction or conclusion.

Line 290-292: I don't clearly see the distributions "moving left" throughout the snow season in Figure 4; are zero values removed? This might be more effective if quantified with the skewness and kurtosis of the data (retaining zeros) to illustrate how the distribution changes throughout the snow season. I also wonder if these figures would be more effective if multiple dates were combined in one panel, with semi-transparent fill to facilitate comparison of the distributions throughout the snow season.

Line 293-296: The bimodality is an interesting observation, although it's not clear to me that this is enough bimodality to be significant. Could you run a statistical test for bimodality in the distribution?

Line 313-317: This seems like either methods or motivation, not results.

Figure 7: This is a suggestion only, but I wonder if this figure would be more effective with elevation on the y-axis, volume on the x-axis, and color denoting the dates of observations. That might facilitate an understanding of the hypsometric effects on snow distributions.

Line 323-325: The finding that higher elevation bands have slower snowmelt is surprising, and I think contradicts the "slower snowmelt in a warmer world" idea (Musselman et al., 2017). This needs to be explained more thoroughly.

Line 337: Consider "delayed melt timing" instead of "slower melt rates"

Line 361-363: Why would the fact that there's a transition from dry to very snowy imply that these years are particularly beneficial for illustrating the spatial patterns of peak SWE? The emphasis on a transition here implies that there was a lag effect of the snow drought year, but I think you just mean that there's a lot of variability in these years, right? Again, be cautious of describing 2017 as one of the snowiest on record.

Line 400-401: I would advocate for showing this data, at least as supplemental.

Line 448: Unaccounted for in what sense? If we assume that basin-wide maximum SWE is the total volume of water available for seasonal streamflow forecasting?

Figures 12-14: The maps here might be more legible if they used a continuous color scale, since they ultimately represent continuous variables. The only challenge is that a separate color scale (or other aesthetic) would then be required for the aspect plots and histogram.

Line 515: This is the first time I'm understanding that the authors see the analysis of SWE, rather than depth, as a major unique contribution. If that's the case, why not compare the results of the SWE analysis to what would be obtained using depth? It's not clear to me that they would be qualitatively different. The entire following paragraph (Lines 521-537) would be more relevant in that case; as is, it's not clear to me that the depth vs SWE difference is particularly important.

Line 547: I think the "decaying function" needs to be defined and analyzed more clearly for this claim to be made.

Line 559-563: The relationship of these mechanisms (less SWE at the highest elevation) to the findings in lines 563-566 seems unnecessarily complicated. Figure 8 suggests that these mechanisms are often (except in 2014) relevant at the very highest elevations only. I don't really think Figure 7 is relevant to this paragraph, as the entanglement of SWE depth and hypsometry makes it difficult to assess the underlying mechanisms.

Line 570-577: This previous work seems only partially relevant – how are the details important, besides the fact that not much previous work has analyzed snow-hypsometry relationships?

Line 582-590: I'm not convinced that the analysis of different snowmelt timing across elevations is that unique, although the use of lidar data for it may be. I think a better contribution here is the analysis of how these elevational differences vary across years.

Line 595: But does the Comola et al. (2015) result depend on lidar data? This sounds like a study that could be done with a model only; the need for the spatially distributed data isn't clear.

Line 614-615: Could you provide evidence that the named entities are exploring new and extended deployments of airborne and satellite snow measurement programs?

Line 630: As written, this seems to presume that the only viable alternative to one ASO acquisition is more ASO acquisitions. But what about modeling, or analyzing data from other sources? E.g. see Raleigh et al. (2025).

Line 637-639: It's good to see the time intervals between flights acknowledged here, but as described in the major comments, I think this uncertainty needs to be addressed much more directly.

Line 640-641: I'm having trouble following the structure here. It seems that the text in roughly lines 620-630 was already aimed at discussing the question of the limitations of a single acquisition.

Line 660: The prior section was conceptually repetitive, but this point about CA DWR is directly repetitive with Lines 630.

Line 676-677: This study doesn't really focus on the findings of a high-resolution snow model, although I recognize that one is used in the SWE estimation. If the focus was on the findings of a high-resolution snow model, you could look at the basin peak SWE and date of peak SWE estimated by the model, and avoid all the problems with the intermittent lidar acquisitions.

Line 680: As above, I'm not sure it's important that there was a progression between years when there's no interannual hysteresis.

Line 703-705: I'm not convinced that the results accomplish all these things (reservoir operations, remote sensing design, ground survey design). Could you say specifically how it supports these outcomes?

**References**

Herbert, J. N., Raleigh, M. S., & Small, E. E. (2024). Reanalyzing the spatial representativeness of snow depth at automated monitoring stations using airborne lidar data. *The Cryosphere*, *18*(8), 3495–3512. https://doi.org/10.5194/tc-18-3495-2024

Koshkin, A., & Marshall, A. (2025). Airborne Lidar and Machine Learning Reveal Decreased Snow Depth in Burned Forests. *EGUsphere*, 1–25. https://doi.org/10.5194/egusphere-2025-4081

Marshall, A. M., Abatzoglou, J. T., Rahimi, S., Lettenmaier, D. P., & Hall, A. (2024). California's 2023 snow deluge: Contextualizing an extreme snow year against future climate change. *Proceedings of the National Academy of Sciences*, *121*(20), e2320600121. https://doi.org/10.1073/pnas.2320600121

Marshall, A. M., Abatzoglou, J. T., Link, T. E., & Tennant, C. J. (2019). Projected Changes in Interannual Variability of Peak Snowpack Amount and Timing in the Western United States. *Geophysical Research Letters*, *46*(15), 8882–8892. https://doi.org/10.1029/2019GL083770

Montoya, E. L., Dozier, J., & Meiring, W. (2014). Biases of April 1 snow water equivalent records in the Sierra Nevada and their associations with large-scale climate indices. *Geophysical Research Letters*, *41*(16), 5912–5918. https://doi.org/10.1002/2014GL060588

Musselman, K. N., Clark, M. P., Liu, C., Ikeda, K., & Rasmussen, R. (2017). Slower snowmelt in a warmer world. *Nature Climate Change*, *7*(3), 214–219. https://doi.org/10.1038/nclimate3225

Raleigh, M. S., Small, E. E., Bair, E. H., Wobus, C., & Rittger, K. (2025). Snow monitoring at strategic locations improves water supply forecasting more than basin-wide mapping. *Communications Earth & Environment*, *6*(1), 665. https://doi.org/10.1038/s43247-025-02660-z